# Facial Action Unit Recognition by Prior and Adaptive Attention

Zhiwen Shao [1,2], Yong Zhou [1,2,*], Hancheng Zhu [1,2], Wen-Liang Du [1,2], Rui Yao [1,2] and Hao Chen [3]

1   School of Computer Science and Technology, China University of Mining and Technology,
    Xuzhou 221116, China
2   Engineering Research Center of Mine Digitization, Ministry of Education of the People's Republic of China,
    Xuzhou 221116, China
3   Xuzhou Guanglian Technology Co., Ltd., Xuzhou 221116, China
*   Correspondence: yzhou@cumt.edu.cn

**Abstract:** Facial action unit (AU) recognition remains a challenging task, due to the subtlety and non-rigidity of AUs. A typical solution is to localize the correlated regions of each AU. Current works often predefine the region of interest (ROI) of each AU via prior knowledge, or try to capture the ROI only by the supervision of AU recognition during training. However, the predefinition often neglects important regions, while the supervision is insufficient to precisely localize ROIs. In this paper, we propose a novel AU recognition method by prior and adaptive attention. Specifically, we predefine a mask for each AU, in which the locations farther away from the AU centers specified by prior knowledge have lower weights. A learnable parameter is adopted to control the importance of different locations. Then, we element-wise multiply the mask by a learnable attention map, and use the new attention map to extract the AU-related feature, in which AU recognition can supervise the adaptive learning of a new attention map. Experimental results show that our method (i) outperforms the state-of-the-art AU recognition approaches on challenging benchmark datasets, and (ii) can accurately reason the regional attention distribution of each AU by combining the advantages of both the predefinition and the supervision.

**Keywords:** facial AU recognition; prior knowledge; adaptive attention

## 1. Introduction

Facial action unit (AU) recognition involves the prediction for occurrence or non-occurrence of each AU, which is an important task in the communities of computer vision and affective computing [1–4]. As defined in the facial action coding system (FACS) [5,6], each AU is a local facial action with one or more atomic muscle actions. Due to the subtlety and non-rigidity, the appearance of AUs are diversely changed across persons and expressions. For instance, as shown in Figure 1, AU 1 (inner brow raiser), AU 2 (outer brow raiser), and AU 4 (brow lowerer) occur in brow regions with overlaps, in which it is difficult to distinguish each AU from the fused appearance. In the literature, facial AU recognition is still a challenging task.

Considering AUs appear in local facial regions, one intuitive solution is to localize the correlated regions so as to extract features for AU recognition. Since the locations of AU centers can be specified by correlated facial landmarks via prior knowledge, Li et al. [2,7] predefined a regional attention map for each AU, in which a position with a farther distance to the AU centers is given a lower attention value. However, different AUs share the same attention distribution, which ignores the divergences across AUs. Furthermore, correlated landmarks only can determine the central locations of AUs, while a few potentially correlated regions very far away from the centers are neglected.

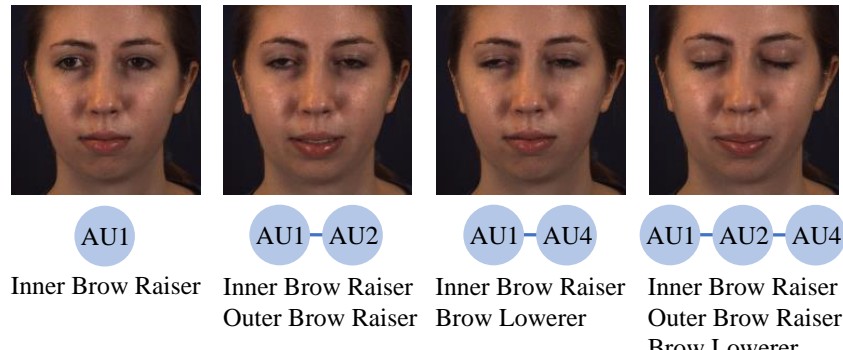

**Figure 1.** Example images for different action units (AUs) with overlapped regions. If the regions of multiple AUs have overlaps, the appearance of each AU fuses for a new combined appearance.

As deep neural networks have a self-attention mechanism [8] during training, Shao et al. [9] only resorted to the supervision of AU recognition to capture correlated regions of AUs. In this approach, some irrelevant regions are included since AUs are subtle and do not have distinct contours. Shao et al. [3,10] proposed adaptively modifying the predefined attention map of each AU, which is a pioneering work of combining predefined attention and supervised attention. However, directly convoluting on a predefined attention map only works as smoothing, in which positions outside of the predefined region of interest (ROI) obtain similar attention weights and thus are regarded as similar importance. In this way, correlated regions very distant to the AU centers are still not emphasized.

To tackle the above issues, we develop a novel facial AU recognition method named **PAA** by prior and adaptive attention. In particular, we first predefine a mask for each AU by assigning lower weights to the positions farther away from the predefined AU centers. Since the regions outside the predefined ROI should not be neglected, we use a learnable parameter to adaptively control the importance of different positions. Then, we element-wise multiply the mask by a learnable attention map, and employ the new attention map to extract AU-related features. In this process, the new attention map is adaptively learned under the guidance of AU recognition. By integrating the advantages of the predefinition and the supervision, our method can precisely capture correlated regions of each AU, in which correlated locations are included and uncorrelated locations are discarded.

This paper has three main contributions:

- We propose to combine the constraint of prior knowledge and the supervision of AU recognition to adaptively learn the regional attention distribution of each AU.
- We propose a learnable parameter to adaptively control the importance of different positions in the predefined mask, which is beneficial for choosing an appropriate constraint of prior knowledge.
- We conduct extensive experiments on challenging benchmark datasets, which demonstrate that our method outperforms the state-of-the-art AU recognition approaches, and can precisely learn the attention map of each AU.

## 2. Related Works

In this section, we review other approaches that are strongly relevant to our work, including facial landmark-aided AU recognition approaches and attention learning-based AU recognition approaches.

## 2.1. Facial Landmark-Aided AU Recognition

Since facial landmarks have prior location relationships with AUs, landmarks can help to learn AU-related features. Benitez-Quiroz et al. [11] integrated the geometry and local texture feature for AU recognition, in which the geometry feature contains the normalized distances among landmarks as well as the angles of Delaunay mask constructed by landmarks. Zhao et al. [12] extracted scale-invariant feature transform (SIFT) [13] features from local facial regions centered at relevant landmarks as AU-related features.

Facial landmarks can also facilitate AU recognition in other ways. Niu et al. [14] relied on landmarks to construct a shape regularization to AU recognition. Ma et al. [15] introduced typical object detection tasks into AU recognition by employing landmarks to define bounding boxes for AUs, in which each AU is predicted to occur in which bounding box. If one AU is absent, it should be predicted as non-occurrence for all bounding boxes.

These approaches all demonstrate the contribution of landmarks to AU recognition. In this paper, we use landmarks to predefine a regional mask with a learnable control parameter for each AU.

## 2.2. Attention Learning-Based AU Recognition

Considering AUs are subtle and have no distinct contour and texture, it is impracticable to manually annotate their regional attention distribution. An intuitive solution is to use the prior knowledge for attention predefinition. Li et al. [2,7] generated an attention map for each AU by predefining two Gaussian distributions centered around the two AU centers due to the symmetry, in which a position farther away from the centers has a smaller attention weight. Different AUs have identical attention distributions, which ignore the differences among AUs. Furthermore, attention predefinition cannot highlight potentially correlated regions far away from the predefined ROI.

As important region beyond the predefined ROI may be neglected, Shao et al. [9] directly learned the attention map of each AU without the prior knowledge, while Shao et al. [3,10] modified the predefined attention map under the supervision of AU recognition. However, the attention distribution learned in [9] contains quite a few uncorrelated locations, and the attention modification in [3,10] seems to be the smoothing of predefintion and still cannot emphasize correlated locations very distant to the predefined ROI. On the contrary, our approach can include both strongly correlated positions near the AU centers and weakly correlated positions scattered on the global face.

## 3. PAA for Facial AU Recognition

### 3.1. Overview

Our main goal is to predict the occurrence probability of total $m$ AUs for an input image: $\hat{\mathbf{p}} = (\hat{p}_1, \cdots, \hat{p}_m)$. The structure of our network is illustrated in Figure 2. Specifically, two hierarchical and multi-scale region layers [3,10] with each, followed by a max-pooling layer, are firstly used to extract a multi-scale feature, which is beneficial for adapting to AUs with diverse sizes in different local facial regions. Then, each AU has one branch to predict the occurrence probability, in which the predefined mask $\mathbf{M}_i$, as the prior knowledge, constrains the learning of new attention map $\widehat{\mathbf{M}}_i^{(1)}$ during training. In our framework, both the prior knowledge and the AU recognition guidance are exploited to learn the regional attention distribution of each AU.

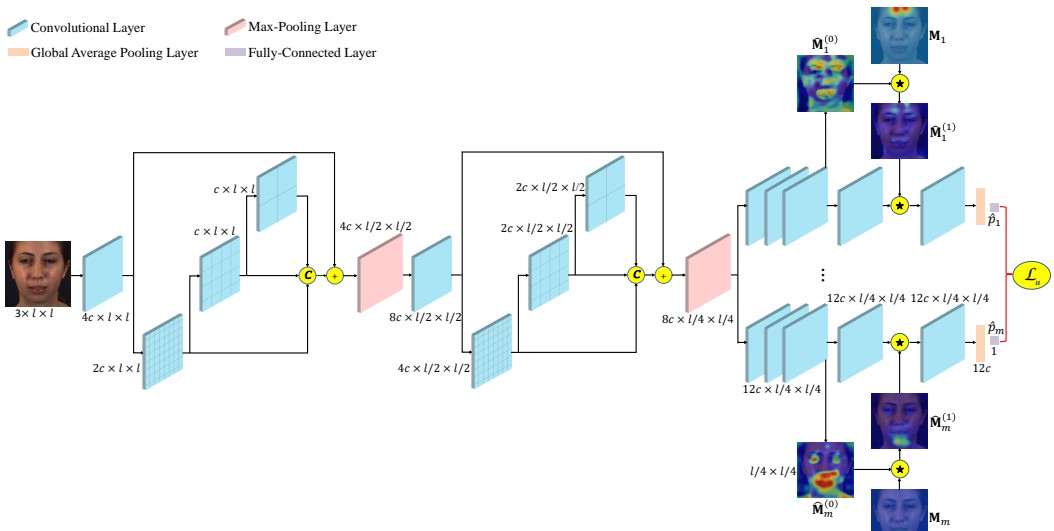

**Figure 2.** The overview of our PAA framework. An input image firstly goes through two hierarchical and multi-scale region layers [3,10], each of which is followed by a max-pooling layer. Then, $m$ branches are used to predict AU occurrence probabilities, in which the learned attention map $\widehat{\mathbf{M}}_i^{(0)}$ is element-wise multiplied by the predefined mask $\mathbf{M}_i$ to obtain the new attention map $\widehat{\mathbf{M}}_i^{(1)}$ for the $i$-th AU. We overlay the attention maps as well as the masks on the input image for a better view. "$\star$" refers to element-wise multiplication. The expression $c \times l \times l$ refers to the layer dimensions are $c$, $l$, and $l$, respectively.

### 3.2. Constraint of Attention Predefinition

In the branch of the $i$-th AU, three convolutional layers are firstly adopted, where $i = 1, \cdots, m$. Then, a convolutional layer with one channel is used to learn the attention map $\widehat{\mathbf{M}}_i^{(0)}$. According to the prior knowledge, the central locations of AUs can be determined by correlated facial landmarks [2,3], as illustrated in Figure 3. To exploit the prior knowledge, we predefine a mask $\mathbf{M}_i$ for the $i$-th AU.

| AU | Location |
|---|---|
| ● 1 (Inner brow raiser) | 1/2 scale above inner brow |
| ● 2 (Outer brow raiser) | 1/3 scale above outer brow |
| ● 4 (Brow lowerer) | 1/3 scale below brow center |
| ● 6 (Cheek raiser) | 1 scale below eye bottom |
| ● 7 (Lid tightener) | Eye |
| ● 9 (Nose wrinkler) | 1/2 scale above nose bottom |
| ● 10 (Upper lip raiser) | Upper lip center |
| 12 (Lip corner puller)<br>● 14 (Dimpler)<br>15 (Lip corner depressor) | Lip corner |
| ● 17 (Chin raiser)<br>26 (Jaw drop) | 1/2 scale below lip |
| 23 (Lip tightener)<br>● 24 (Lip pressor)<br>25 (Lips part) | Lip center |

**Figure 3.** Definition of the central locations of 15 popular AUs for a normalized face with eye centers on the same horizontal level [2,3], in which "scale" denotes the distance between two inner eye corners. Each AU has two symmetric centers specified by two correlated facial landmarks, in which landmarks are in white and AU centers are in other colors.

Since the $i$-th AU has two centers $(\bar{a}_{i(1)}, \bar{b}_{i(1)})$ and $(\bar{a}_{i(2)}, \bar{b}_{i(2)})$, we take the predefined mask $\widetilde{\mathbf{M}}_{i(1)}$ of the first center as an example. In particular, we use a Gaussian distribution with a standard deviation $\delta$ centered around the location $(\bar{a}_{i(1)}, \bar{b}_{i(1)})$ to compute the value at each location $(a, b)$:

$$\widetilde{M}_{iab(1)} = \exp\left(-\frac{(a - \bar{a}_{i(1)})^2 + (b - \bar{b}_{i(1)})^2}{2\delta^2}\right). \tag{1}$$

We next incorporate $\widetilde{\mathbf{M}}_{i(1)}$ and $\widetilde{\mathbf{M}}_{i(2)}$ by choosing the larger value at each location:

$$\widetilde{M}_{iab} = \max(\widetilde{M}_{iab(1)}, \widetilde{M}_{iab(2)}) \in (0, 1]. \tag{2}$$

In $\widetilde{\mathbf{M}}_i$, the positions with values significantly larger than zero constitute the ROI of the $i$-th AU, while other approximately zero-valued positions are ignored. However, the positions beyond the predefined ROI that we do not want are completely discarded during the constraint for attention learning. We introduce a learnable—instead of fixed—control parameter $\epsilon_i$ to give appropriate importance to the positions outside of the ROI:

$$M_{iab} = \frac{\widetilde{M}_{iab} + \epsilon_i}{1 + \epsilon_i} \in (0, 1], \tag{3}$$

where $\epsilon_i \geq 0$ and a larger $\epsilon_i$ give larger importance to the positions beyond the predefined ROI. Note that the relative size between different positions in $\widetilde{\mathbf{M}}_i$ is unchanged, and different AUs have independent control parameters. As illustrated in Figure 2, $\mathbf{M}_1$ and $\mathbf{M}_m$ are adaptively learned with different attention distributions.

Supervision of AU Recognition

After obtaining the predefined mask $\mathbf{M}_i$, we generate the new attention map $\widehat{\mathbf{M}}_i^{(1)}$ by element-wise multiplying $\mathbf{M}_i$ by $\widehat{\mathbf{M}}_i^{(0)}$:

$$\widehat{\mathbf{M}}_i^{(1)} = \mathbf{M}_i \star \widehat{\mathbf{M}}_i^{(0)}. \tag{4}$$

Considering that deep neural networks have a self-attention mechanism [8], we exploit AU recognition to guide the learning of $\widehat{\mathbf{M}}_i^{(1)}$.

Specifically, as shown in Figure 2, we element-wise multiply $\widehat{\mathbf{M}}_i^{(1)}$ with the fourth convolutional channel map to emphasize AU-related features. Then, another convolutional layer, as well as a global average pooling layer [16], are adopted to extract the AU feature with the size of $12c$. Finally, we predict the occurrence probability $\hat{p}_i$ of the $i$-th AU by using a fully-connected layer with one dimension followed by a Sigmoid function, and define the AU recognition loss as:

$$\mathcal{L}_u = -\sum_{i=1}^{m} w_i[v_i p_i \log \hat{p}_i + (1 - p_i) \log(1 - \hat{p}_i)], \tag{5}$$

where a weighting strategy is employed, and $p_i$, $w_i$, and $v_i$ denote the ground-truth occurrence probability, the weight, and the weight for occurrence of the $i$-th AU, respectively. There are two types of data imbalance issues [17] in most existing AU datasets [18–20]: inter-AU data imbalance that different AUs have different occurrence rates, and intra-AU data imbalance that AUs often have smaller occurrence rates than non-occurrence rates. To alleviate these data imbalance issues, $w_i$ and $v_i$ are defined as:

$$w_i = \frac{n}{n_i^{occ}} \Big/ \sum_{k=1}^{m} \frac{n}{n_k^{occ}}, \quad v_i = \frac{n - n_i^{occ}}{n_i^{occ}}, \tag{6}$$

where $n_i^{occ}/n$ is the occurrence rate of the $i$-th AU, and $n_i^{occ}$ and $n$ denote the number of images appearing in the $i$-th AU and the number of all images in the training set, respectively.

By the constraint of attention predefinition and the guidance of AU recognition, the adaptively learned AU attention map $\widehat{\mathbf{M}}_i^{(1)}$ can capture both strongly relevant regions predefined by prior knowledge as well as scattered relevant regions on the global face. In this case, our AU recognition method can work well under the subtlety and non-rigidity of AUs due to the accuracy of AU-related features.

## 4. Experiments

### 4.1. Datasets and Settings

#### 4.1.1. Datasets

In this paper, we evaluate our PAA on three popular benchmark datasets. Besides AU annotations, each dataset is also annotated with facial landmarks.

- **Binghamton-Pittsburgh 4D (BP4D)** [18] includes 41 subjects, including 23 women and 18 men, in which 328 videos with about 140,000 frames are captured in total by placing each subject into 8 sessions. Each frame is labeled with the AUs of occurrence or non-occurrence. Similar to the previous approaches [1–3], we conduct subject-exclusive three-fold cross-validation with two folds for training and the remaining one for testing on 12 AUs. Our method uses the same partitions of subjects as the previous works [1–3].

- **Denver Intensity of Spontaneous Facial Action (DISFA)** [19] contains 12 women and 15 men, in which each subject is recorded by a video with 4845 frames. Each frame is labeled with AU intensities ranging from 0 to 5. Following the previous methods [1–3], we treat an AU as occurrence if its intensity is equal or larger than two, and treat it as non-occurrence otherwise. We conduct subject-exclusive three-fold cross-validation on eight AUs. Our method uses the same partitions of subjects as the previous works [1–3].

- **Sayette Group Formation Task (GFT)** [20] includes 96 subjects with each subject captured by one video, whose images are more challenging than BP4D and DISFA due to unscripted interactions in 32 three-subject teams. Each frame is labeled with AU occurrences. We adopt the official training and testing partitions of subjects [20], in which about 108,000 frames of 78 subjects are used for training, and about 24,600 frames of 18 subjects are used for testing.

#### 4.1.2. Implementation Details

Our PAA is implemented via PyTorch [21], in which each convolutional layer adopts $3 \times 3$ convolutional filters with a stride of 1, a padding of 1, and each max-pooling layer processes $2 \times 2$ spatial fields with a stride of 2. We normalize each image to $3 \times 200 \times 200$ by similarity transformation, and randomly crop the normalized image to $3 \times l \times l$ with a random horizontal flip. The image size $l$, the network parameter $c$, as well as the standard deviation $\delta$ in Equation (1) are set to 176, 8, and 3, respectively.

Similar to J$\hat{A}$A-Net [3], we employ the stochastic gradient descent (SGD) solver with a Nesterov momentum [22] of 0.9, a weight decay of 0.0005, and a mini-batch size of 8 to train PAA with 12 epochs. The learning rate is initialized to be 0.006 and is multiplied by 0.3 at every 2 epochs during training. Following the settings in [1–3], we use the parameters of the well-trained model on BP4D for initialization when training on DISFA.

#### 4.1.3. Evaluation Metrics

We evaluate methods via a popular metric frame-based F1-score (F1-frame):

$$F1 = \frac{2PR}{P + R},\tag{7}$$

where $P$ denotes the precision, and $R$ denotes the recall. We also report the average results of the F1-frame over all AUs (shortly written as Avg). We omit "%" in all the F1-frame results for simplicity in the experimental results.

### 4.2. Comparison with State-of-the-Art Methods

In this section, we compare our PAA approach with state-of-the-art AU recognition methods, including LSVM [23], APL [24], JPML [12], AlexNet [25], DRML [1], EAC-Net [2], DSIN [26], CMS [27], LP-Net [14], ARL [9], SRERL [28], AU R-CNN [15], TCAE [29], AU-GCN [30], Ertugrul et al. [31], JÂA-Net [3], UGN-B [32], HMP-PS [33], and GeoCNN [34]. Notice that these works often adopt external training data, while our PAA uses the benchmark dataset only. Specifically, EAC-Net, SRERL, AU R-CNN, UGN-B, HMP-PS, and GeoCNN use pre-trained ImageNet models [35,36], CMS employs external thermal images, LP-Net pre-trains on a face recognition dataset [37], and GeoCNN utilizes a pre-trained 3D morphable model (3DMM) [38,39]. Several related works such as R-T1 [7] are not compared since they process a sequence of frames instead of a single frame.

#### 4.2.1. Evaluation on BP4D

Table 1 shows the results of different methods on the BP4D benchmark. We can observe that our PAA performs better than most of the previous works, especially for the average F1-frame. Compared to other methods using external training data, such as AU R-CNN and UGN-B, PAA uses benchmark training images only, while achieving better performance. Although GeoCNN is slightly better than our method, it relies on a pre-trained 3DMM to obtain additional 3D manifold information to facilitate AU recognition.

**Table 1.** F1-frame results on Binghamton-Pittsburgh 4D (BP4D) [18]. The results of LSVM [23] and JPML [12] are from [1], and those of other previous methods are reported in their original papers. The best results of each AU, as well as the average across methods, are shown in bold. Our PAA method performs better than most of the previous works.

| AU | 1 | 2 | 4 | 6 | 7 | 10 | 12 | 14 | 15 | 17 | 23 | 24 | Avg |
|---|---|---|---|---|---|---|---|---|---|---|---|---|---|
| LSVM [23] | 23.2 | 22.8 | 23.1 | 27.2 | 47.1 | 77.2 | 63.7 | 64.3 | 18.4 | 33.0 | 19.4 | 20.7 | 35.3 |
| JPML [12] | 32.6 | 25.6 | 37.4 | 42.3 | 50.5 | 72.2 | 74.1 | 65.7 | 38.1 | 40.0 | 30.4 | 42.3 | 45.9 |
| DRML [1] | 36.4 | 41.8 | 43.0 | 55.0 | 67.0 | 66.3 | 65.8 | 54.1 | 33.2 | 48.0 | 31.7 | 30.0 | 48.3 |
| EAC-Net [2] | 39.0 | 35.2 | 48.6 | 76.1 | 72.9 | 81.9 | 86.2 | 58.8 | 37.5 | 59.1 | 35.9 | 35.8 | 55.9 |
| DSIN [26] | 51.7 | 40.4 | 56.0 | 76.1 | 73.5 | 79.9 | 85.4 | 62.7 | 37.3 | 62.9 | 38.8 | 41.6 | 58.9 |
| CMS [27] | 49.1 | 44.1 | 50.3 | 79.2 | 74.7 | 80.9 | **88.3** | 63.9 | 44.4 | 60.3 | 41.4 | 51.2 | 60.6 |
| LP-Net [14] | 43.4 | 38.0 | 54.2 | 77.1 | 76.7 | 83.8 | 87.2 | 63.3 | 45.3 | 60.5 | 48.1 | 54.2 | 61.0 |
| ARL [9] | 45.8 | 39.8 | 55.1 | 75.7 | 77.2 | 82.3 | 86.6 | 58.8 | 47.6 | 62.1 | 47.4 | 55.4 | 61.1 |
| SRERL [28] | 46.9 | 45.3 | 55.6 | 77.1 | 78.4 | 83.5 | 87.6 | 60.6 | 52.2 | 63.9 | 47.1 | 53.3 | 62.9 |
| AU R-CNN [15] | 50.2 | 43.7 | 57.0 | 78.5 | 78.5 | 82.6 | 87.0 | **67.7** | 49.1 | 62.4 | 50.4 | 49.3 | 63.0 |
| AU-GCN [30] | 46.8 | 38.5 | **60.1** | **80.1** | **79.5** | **84.8** | 88.0 | 67.3 | 52.0 | 63.2 | 40.9 | 52.8 | 62.8 |
| JÂA-Net [3] | 53.8 | **47.8** | 58.2 | 78.5 | 75.8 | 82.7 | 88.2 | 63.7 | 43.3 | 61.8 | 45.6 | 49.9 | 62.4 |
| UGN-B [32] | **54.2** | 46.4 | 56.8 | 76.2 | 76.7 | 82.4 | 86.1 | 64.7 | 51.2 | 63.1 | 48.5 | 53.6 | 63.3 |
| HMP-PS [33] | 53.1 | 46.1 | 56.0 | 76.5 | 76.9 | 82.1 | 86.4 | 64.8 | 51.5 | 63.0 | 49.9 | 54.5 | 63.4 |
| GeoCNN [34] | 48.4 | 44.2 | 59.9 | 78.4 | 75.6 | 83.6 | 86.7 | 65.0 | **53.0** | **64.7** | 49.5 | 54.1 | **63.6** |
| **PAA** | 50.1 | 47.7 | 55.0 | 74.0 | 78.9 | 82.2 | 87.2 | 63.8 | 51.4 | 62.4 | **52.1** | **55.8** | 63.4 |

#### 4.2.2. Evaluation on DISFA

Table 2 shows the results on DISFA, from which we can see that our PAA achieves competitive performance. It can also be found that many methods such as AU-GCN exhibit more fluctuated results across AUs on DISFA than on BP4D, and work well on BP4D, but show poor results on DISFA. This is because DISFA is more challenging with a more severe data imbalance problem than BP4D. In this case, our PAA achieves a more stable performance among different AUs than most of the previous works, and performs consistently well on BP4D and DISFA with 63.4 and 62.9 average F1-frame results, respectively.

**Table 2.** F1-frame results on Denver Intensity of Spontaneous Facial Action (DISFA) [19]. The results of LSVM [23] and APL [24] are from [1], and those of other previous methods are reported in their original papers. The best results of each AU, as well as the average across methods, are shown in bold. Our PAA method achieves competitive performance, and achieves a more stable performance among different AUs than most of the previous works.

| AU | 1 | 2 | 4 | 6 | 9 | 12 | 25 | 26 | Avg |
|---|---|---|---|---|---|---|---|---|---|
| LSVM [23] | 10.8 | 10.0 | 21.8 | 15.7 | 11.5 | 70.4 | 12.0 | 22.1 | 21.8 |
| APL [24] | 11.4 | 12.0 | 30.1 | 12.4 | 10.1 | 65.9 | 21.4 | 26.9 | 23.8 |
| DRML [1] | 17.3 | 17.7 | 37.4 | 29.0 | 10.7 | 37.7 | 38.5 | 20.1 | 26.7 |
| EAC-Net [2] | 41.5 | 26.4 | 66.4 | 50.7 | 8.5 | **89.3** | 88.9 | 15.6 | 48.5 |
| DSIN [26] | 42.4 | 39.0 | 68.4 | 28.6 | 46.8 | 70.8 | 90.4 | 42.2 | 53.6 |
| CMS [27] | 40.2 | 44.3 | 53.2 | 57.1 | 50.3 | 73.5 | 81.1 | 59.7 | 57.4 |
| LP-Net [14] | 29.9 | 24.7 | **72.7** | 46.8 | 49.6 | 72.9 | 93.8 | 65.0 | 56.9 |
| ARL [9] | 43.9 | 42.1 | 63.6 | 41.8 | 40.0 | 76.2 | 95.2 | 66.8 | 58.7 |
| SRERL [28] | 45.7 | 47.8 | 59.6 | 47.1 | 45.6 | 73.5 | 84.3 | 43.6 | 55.9 |
| AU R-CNN [15] | 32.1 | 25.9 | 59.8 | 55.3 | 39.8 | 67.7 | 77.4 | 52.6 | 51.3 |
| AU-GCN [30] | 32.3 | 19.5 | 55.7 | **57.9** | **61.4** | 62.7 | 90.9 | 60.0 | 55.0 |
| JÂA-Net [3] | 62.4 | 60.7 | 67.1 | 41.1 | 45.1 | 73.5 | 90.9 | 67.4 | **63.5** |
| UGN-B [32] | 43.3 | 48.1 | 63.4 | 49.5 | 48.2 | 72.9 | 90.8 | 59.0 | 60.0 |
| HMP-PS [33] | 38.0 | 45.9 | 65.2 | 50.9 | 50.8 | 76.0 | 93.3 | 67.6 | 61.0 |
| GeoCNN [34] | **65.5** | **65.8** | 67.2 | 48.6 | 51.4 | 72.6 | 80.9 | 44.9 | 62.1 |
| **PAA** | 56.1 | 57.0 | 59.0 | 39.7 | 49.4 | 74.6 | **95.6** | **71.9** | 62.9 |

### 4.2.3. Evaluation of GFT

The comparison results on GFT are presented in Table 3. It can be observed that our PAA outperforms all other approaches. Notice that GFT images are often in large poses, which are more challenging than BP4D and DISFA images with near-frontal poses. In this case, PAA still achieves good performance with the highest average F1-frame of 55.8.

**Table 3.** F1-frame results of Sayette Group Formation Task (GFT) [20]. The results of LSVM [23] and AlexNet [25] are from [20], those of EAC-Net [2] and ARL [9] are from [3], and those of other previous methods are reported in their original papers. The best results of each AU, as well as the average across methods, are shown in bold. Our PAA method outperforms all the other approaches.

| AU | 1 | 2 | 4 | 6 | 10 | 12 | 14 | 15 | 23 | 24 | Avg |
|---|---|---|---|---|---|---|---|---|---|---|---|
| LSVM [23] | 38 | 32 | 13 | 67 | 64 | 78 | 15 | 29 | 49 | 44 | 42.9 |
| AlexNet [25] | 44 | 46 | 2 | 73 | 72 | 82 | 5 | 19 | 43 | 42 | 42.8 |
| EAC-Net [2] | 15.5 | **56.6** | 0.1 | **81.0** | 76.1 | 84.0 | 0.1 | 38.5 | 57.8 | **51.2** | 46.1 |
| TCAE [29] | 43.9 | 49.5 | 6.3 | 71.0 | 76.2 | 79.5 | 10.7 | 28.5 | 34.5 | 41.7 | 44.2 |
| ARL [9] | 51.9 | 45.9 | 13.7 | 79.2 | 75.5 | 82.8 | 0.1 | 44.9 | **59.2** | 47.5 | 50.1 |
| Ertugrul et al. [31] | 43.7 | 44.9 | **19.8** | 74.6 | **76.5** | 79.8 | **50.0** | 33.9 | 16.8 | 12.9 | 45.3 |
| JÂA-Net [3] | 46.5 | 49.3 | 19.2 | 79.0 | 75.0 | **84.8** | 44.1 | 33.5 | 54.9 | 50.7 | 53.7 |
| **PAA** | **64.6** | 45.4 | 9.8 | 77.9 | 74.8 | 82.8 | 45.4 | **53.3** | 58.9 | 45.0 | **55.8** |

### 4.3. Ablation Study

In this section, we investigate the effectiveness of each component in our PAA. Table 4 summarizes the structures of different variants of PAA, in which Baseline does not have the structure of learning the attention map $\widehat{\mathbf{M}}_i^{(1)}$ and does not utilize the weighting strategy in Equation (5) with $w_i = 1/m$ and $v_i = 1$. The results of different variants of PAA on the BP4D benchmark are presented in Table 5, which use the same hyperparameters, as detailed in Section 4.1.2.

**Table 4.** The architectures of different variants of our PAA. **HM**: two hierarchical and multi-scale region layers, each of which is followed by a max-pooling layer. **C**: five successive convolutional layers in each AU branch. $W^{(au)}$: weighting strategy in Equation (5). $\widehat{\mathbf{M}}^{(0)}$: attention map $\widehat{\mathbf{M}}_i^{(0)}$ for the $i$-th AU. $\mathbf{M}^{(fix)}$: predefined mask $\mathbf{M}_i$ with fixed $\epsilon_i = 0$ for the $i$-th AU. $\mathbf{M}^{(ada)}$: predefined mask $\mathbf{M}_i$ with adaptively learned $\epsilon_i$ for the $i$-th AU. Baseline does not have the structure of obtaining $\widehat{\mathbf{M}}_i^{(0)}$, $\mathbf{M}_i$, and $\widehat{\mathbf{M}}_i^{(1)}$, and does not utilize the weighting strategy in Equation (5) with $w_i = 1/m$ and $v_i = 1$.

| Method | HM | C | $W^{(au)}$ | $\widehat{\mathbf{M}}^{(0)}$ | $\mathbf{M}^{(fix)}$ | $\mathbf{M}^{(ada)}$ | $\mathcal{L}_u$ |
|---|---|---|---|---|---|---|---|
| Baseline | √ | √ | | | | | √ |
| Baseline+$W^{(au)}$ | √ | √ | √ | | | | √ |
| AA | √ | √ | √ | √ | | | √ |
| PA | √ | √ | √ | | √ | | √ |
| PAA$^{(fix)}$ | √ | √ | √ | √ | √ | | √ |
| **PAA** | √ | √ | √ | √ | | √ | √ |

**Table 5.** F1-frame results for 12 AUs of different variants of PAA on BP4D [18]. The best results of each AU, as well as the average across methods, are shown in bold. The performance is gradually improved after adding the proposed components.

| AU | 1 | 2 | 4 | 6 | 7 | 10 | 12 | 14 | 15 | 17 | 23 | 24 | Avg |
|---|---|---|---|---|---|---|---|---|---|---|---|---|---|
| Baseline | 47.8 | 42.1 | 51.4 | 72.6 | 73.4 | 79.5 | 85.6 | 58.3 | 45.3 | 59.9 | 40.8 | 48.8 | 58.8 |
| Baseline+$W^{(au)}$ | 49.0 | 44.2 | 52.9 | 73.9 | 74.9 | 79.3 | 84.5 | 59.1 | 48.0 | 61.1 | 41.8 | 50.4 | 59.9 |
| AA | **50.5** | 41.8 | **55.0** | 75.1 | 75.6 | 80.4 | 86.0 | 60.6 | 50.0 | 61.1 | 50.5 | 51.3 | 61.5 |
| PA | 48.8 | **50.1** | 48.9 | 75.3 | 77.6 | 81.6 | 85.7 | 62.5 | **52.5** | 61.9 | 44.8 | 53.3 | 61.9 |
| PAA$^{(fix)}$ | 46.7 | 45.6 | 53.9 | **75.5** | **78.9** | 82.0 | 86.9 | 60.2 | 52.0 | 61.6 | 48.2 | 52.5 | 62.0 |
| **PAA** | 50.1 | 47.7 | **55.0** | 74.0 | **78.9** | **82.2** | **87.2** | **63.8** | 51.4 | **62.4** | **52.1** | **55.8** | **63.4** |

### 4.3.1. Weighting Strategy for Suppressing Data Imbalance

We can observe that Baseline+$W^{(au)}$ performs better with the average F1-frame of 59.9 than Baseline. This demonstrates the effectiveness of the introduced weighting strategy by alleviating both inter-AU data imbalance and intra-AU data imbalance.

### 4.3.2. Supervision of AU Recognition for Attention Learning

Besides the structure of Baseline+$W^{(au)}$, AA further adaptively learns the attention map $\widehat{\mathbf{M}}_i^{(0)}$ under the supervision of AU recognition and directly element-wise multiplies $\widehat{\mathbf{M}}_i^{(0)}$ with each channel of the fourth convolutional feature map. We can see that AA significantly improves the average F1-frame to 61.5, which shows that the attention map only under the guidance of AU recognition can already capture much useful AU information.

### 4.3.3. Attention Predefinition

Another variant over Baseline+$W^{(au)}$ is PA, which only uses the prior attention by directly element-wise multiplying the predefined mask $\mathbf{M}_i$ with each channel of the fourth convolutional feature map. Since the guidance of AU recognition is not available, $\epsilon_i = 0$ is fixed for $\mathbf{M}_i$. We can see that PA achieves good performance with the average F1-frame of 61.9. This indicates that the prior knowledge is beneficial for AU recognition by specifying the ROI of each AU. We next further explore the effectiveness of combining prior attention and adaptive attention.

After employing the predefined mask $\mathbf{M}_i$ with the adaptively learned $\epsilon_i$ over AA, our PAA achieves the highest average F1-frame 63.4. To investigate the usefulness of learnable $\epsilon_i$, we implement PAA$^{(fix)}$ by fixing $\epsilon_i = 0$ in Equation (3) for each AU. In this case, the potentially relevant regions beyond the predefined ROI are ignored, in which the average F1-frame is degraded to 62.0. Therefore, the design of adaptive learning for the control parameter is effective since our PAA can adaptively learn which AU has correlated positions beyond the predefined ROI to be emphasized.

### 4.4. Visual Results

In Figure 4, we visualize the attention maps learned by recent attention learning-based methods, EAC-Net [2], JÂA-Net [3] and ARL [9], as well as $\widehat{\mathbf{M}}_i^{(1)}$, $\widehat{\mathbf{M}}_i^{(0)}$, and $\mathbf{M}_i$ for each AU learned by our PAA. Owing to the subtlety and non-rigidity of AUs, different AUs have different appearances, including shapes and sizes, and AU appearances are also varied across persons and expressions. In this case, AUs should have diverse regional attention distributions. For example, the two images in Figure 4 are both in happy expression, while AU 14 co-occurs with AUs 6, 7, 10, 12, and 17 in the first image, and only occurs alone in the second image. It is expected that the same AU of the two images should have different attention maps.

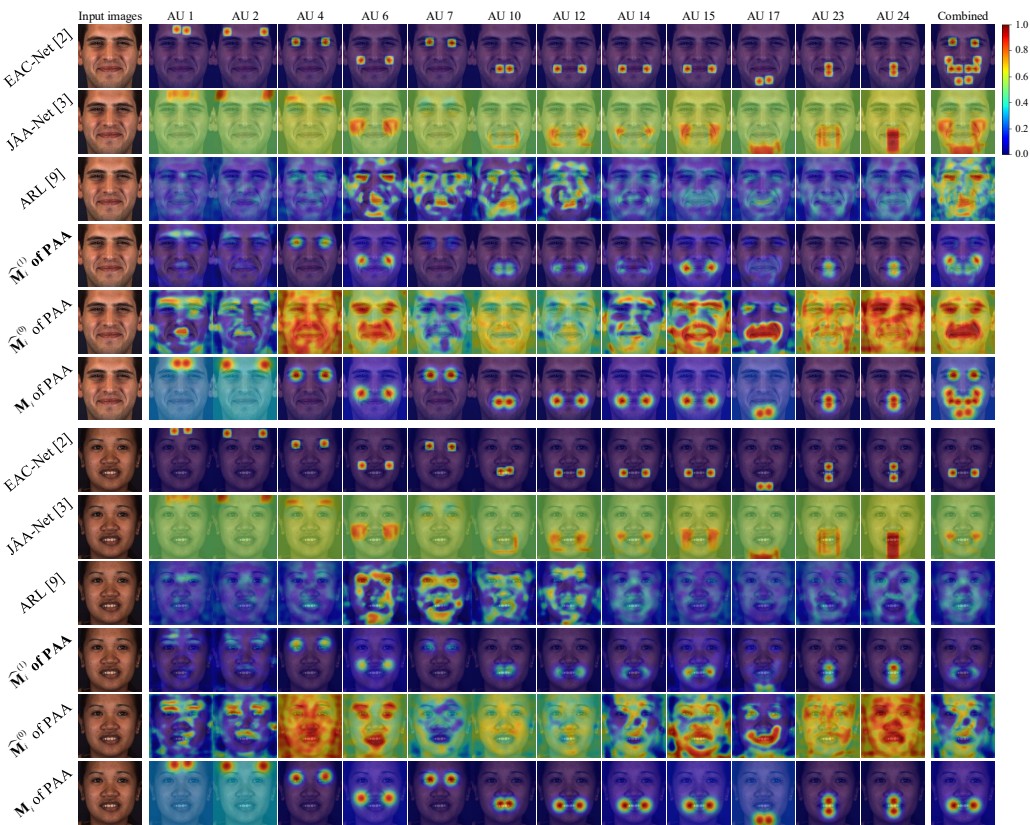

**Figure 4.** Visualization of learned attention maps of different methods, in which AUs 6, 7, 10, 12, 14, and 17 appear in the first image, and AU 14 appears in the second image. Each row lists the attention maps of 12 AUs, as well as the combined attention map of occurred AUs, for the corresponding method. Attention weights from zero to one are visualized using colors from blue to red, and are overlaid on the input images for a better view.

We can observe that different AUs of different images for EAC-Net have the same attention distribution except for different distribution centers, in which the divergences across AUs are ignored and the locations beyond the predefined ROIs are also ignored with zero attention weights. Although JÂA-Net tries to modify the predefined attention maps, it seems that the modification works as the smoothing of the predefinition, in which the positions very distant to the AU centers have smoothed attention weights. Correlated and uncorrelated positions beyond the predefined ROIs are regarded as similar importance, in which the learning of AU features are still often inaccurate. Another solution with a different perspective is ARL, in which the learned attention maps are dense with almost all correlated positions included. However, many potentially uncorrelated positions are mistakenly emphasized.

In contrast with these methods, by combining the advantages of $\mathbf{M}_i$ as the predefinition and $\widehat{\mathbf{M}}_i^{(0)}$ as the supervision for each AU, our PAA can capture both strongly relevant positions specified by landmarks and weakly relevant positions distributed globally in the face. Moreover, we can see that different AUs often have different attention distributions in $\mathbf{M}_i$ since we employ a learnable instead of fixed control parameter for each AU. In this way, we can more adaptively learn the attention weights at different locations of different AUs, especially for those far away from the predefined ROIs. Furthermore, we find that there are quite a few overlaps among the relevant locations in the attention maps of occurred AUs for our PAA. In this case, our combined attention distribution is clean and lies in the highlighted attention range of the predefined combined attention map, which demonstrates that our method can precisely capture the correlated positions of each AU.

**5. Conclusions**

In this paper, we have proposed a novel AU recognition method by prior and adaptive attention, which is beneficial for integrating the advantages of the constraint of prior knowledge and the supervision of AU recognition. We have also proposed a learnable parameter to adaptively control the importance of different positions in the predefined mask of each AU. In this case, we can adaptively learn an appropriate constraint of the prior knowledge.

We have compared our approach against state-of-the-art methods on popular challenging benchmarks, which shows that our approach outperforms most of the previous methods. Besides, we have conducted an ablation study, in which each component in our framework is demonstrated to be contributed to AU recognition. Moreover, the visual results indicate that our approach can accurately reason the regional attention distribution of each AU.

**Author Contributions:** Conceptualization, Z.S. and Y.Z.; methodology, Z.S. and H.Z.; software, Z.S.; validation, W.-L.D. and R.Y.; formal analysis, W.-L.D.; investigation, Z.S.; resources, Z.S.; data curation, H.C.; writing—original draft preparation, Z.S.; writing—review and editing, Z.S., Y.Z. and H.Z.; visualization, Z.S.; supervision, Y.Z.; project administration, Y.Z.; funding acquisition, Y.Z. and H.C. All authors have read and agreed to the published version of the manuscript.

**Funding:** This work was supported by the National Natural Science Foundation of China (No. 62106268), the High-Level Talent Program for Innovation and Entrepreneurship (ShuangChuang Doctor) of Jiangsu Province (No. JSSCBS20211220), and the Talent Program for Deputy General Manager of Science and Technology of Jiangsu Province (No. FZ20220440). It was also partially supported by the National Natural Science Foundation of China (No. 62101555, No. 62002360, and No. 62172417), the Natural Science Foundation of Jiangsu Province (No. BK20201346 and No. BK20210488), the Fundamental Research Funds for the Central Universities (No. 2021QN1072), and the joint project of Guanglian Technology and China University of Mining and Technology.

**Data Availability Statement:** The experiment uses three public AU datasets, including BP4D, DISFA, and GFT. BP4D: http://www.cs.binghamton.edu/~lijun/Research/3DFE/3DFE_Analysis.html (accessed on 30 August 2022); DISFA: http://mohammadmahoor.com/disfa (accessed on 30 August 2022); GFT: https://osf.io/7wcyz (accessed on 30 August 2022).

**Conflicts of Interest:** The authors declare no conflict of interest.

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
