# Peer review of "Facial Action Unit Recognition by Prior and Adaptive Attention"

_electronics, doi:10.3390/electronics11193047_

Round 1

Reviewer 1 Report

Comments: The authors developed a combined strategy with a prior attention mask and an adaptive learnable attention mask to train the PAA model to detect facial action units (AUs), which achieved generally high performance on the benchmark datasets while outperforming other models on certain datasets. The reviewer recognizes this work as sound and well supported by experiments. The overall quality of this manuscript is high and the reviewer enjoys reading it. Some revisions/experiments are suggested to further improve the quality of the manuscript before the manuscript is published.

1.      The baseline model with PA only (with only the prior attention mask) should be evaluated based on the current network structure and hyperparameters. The authors evaluated the AA only model and the reviewer feels that adding the PA only model can make the ablation study comprehensive.

2.      The reviewer suggests the authors perform parallel experiments with randomness (e.g., re-partition the train/val/test for 20-50 times, bootstrap the training data for 20-50 times) to test the robustness of the model performance and test the statistical significance when compared with other models. After all, the authors reported that “It can be observed that our PAA significantly outperforms all the other approaches” (Line 192-193). The reviewer believes that “significantly” should not be used unless the statistical test results support that (p<0.05).

3.      For the ablation study, are the hyperparameters tuned? If possible, the reviewer suggests that the authors report the tuned hyperparameters (learning rate, training epoch, L2 penalty, etc.) for other scholars to reproduce this study more easily. Another reason is that the hyperparameters enabling high performance for the PAA model may be far from optimal for the models in the ablation study. It is unfair to compare the PAA model against the ablated models if the hyperparameters are not tuned.

4.      For discussion: did the author try to develop a distinct model for each AU? Will the performance be better if the first several layers of the neural network are not shared by all AUs?

Reviewer 2 Report

The manuscript is well organized. I have been enjoying reading it. But there are still some problems that need to be dealt with:

All Tables in the paper need more descriptions and more explained.

Author Response

Point 1: All Tables in the paper need more descriptions and more explained.

Response 1: Thank you for your suggestions. In our revised paper, we supplement more descriptions and explanations in each Table.

Reviewer 3 Report

The authors proposed a novel AU recognition method by prior and adaptive attention. The authors also presented a learnable parameter to adaptively control the importance of different positions in the predefined mask of each AU. As reported in the SoA, the authors faced one of the most challenging tasks in the facial expression recognition field. Their methodology in facing this challenge has been excellent, slightly improving the recognition of some of the AUs.

Author Response

Thank you very much for your comments.

Round 2

Reviewer 1 Report

The reviewer thinks that the quality of the article has been improved after the revision and it can be accepted in the current revised format. The reviewer appreciates the response from the authors.

Reviewer 2 Report

Accept